# Paper-Based Enzymatic Electrochemical Sensors for Glucose Determination

**DOI:** 10.3390/s22166232

**Published:** 2022-08-19

**Authors:** Olaya Amor-Gutiérrez, Estefanía Costa-Rama, M. Teresa Fernández-Abedul

**Affiliations:** Department of Physical and Analytical Chemistry, University of Oviedo, 33006 Oviedo, Spain

**Keywords:** paper-based analytical devices, electrochemical biosensors, µPADs, enzymatic sensors, glucose, enzymes

## Abstract

The general objective of Analytical Chemistry, nowadays, is to obtain best-quality information in the shortest time to contribute to the resolution of real problems. In this regard, electrochemical biosensors are interesting alternatives to conventional methods thanks to their great characteristics, both those intrinsically analytical (precision, sensitivity, selectivity, etc.) and those more related to productivity (simplicity, low costs, and fast response, among others). For many years, the scientific community has made continuous progress in improving glucose biosensors, being this analyte the most important in the biosensor market, due to the large amount of people who suffer from *diabetes mellitus*. The sensitivity of the electrochemical techniques combined with the selectivity of the enzymatic methodologies have positioned electrochemical enzymatic sensors as the first option. This review, focusing on the electrochemical determination of glucose using paper-based analytical devices, shows recent approaches in the use of paper as a substrate for low-cost biosensing. General considerations on the principles of enzymatic detection and the design of paper-based analytical devices are given. Finally, the use of paper in enzymatic electrochemical biosensors for glucose detection, including analytical characteristics of the methodologies reported in relevant articles over the last years, is also covered.

## 1. Introduction

Glucose is one of the most important biological compounds in nature. It is a monosaccharide responsible for the generation of most of the energy required for growth and reproduction, due to its important role in the pathway of aerobic and anaerobic respiration in living organisms. In plants and bacteria, it is usually produced from water and carbon dioxide through photosynthesis, and it is condensed forming starch. Glucose can be stored as an energy reserve or used as a substrate for the synthesis of a great range of saccharides, such as sucrose or cellulose. In the food industry, it is used to enhance texture and flavor [1]. However, the role of glucose in society cannot be explained without mentioning the disease *diabetes mellitus*, a disorder derived from high glucose concentration in blood due to different reasons: abnormal insulin secretion, resistance to insulin, or both [2]. Due to these and its strong relationship with obesity problems, glucose monitoring is an important global issue in health control and food safety [3,4].

In fact, glucose was the target of the first enzymatic biosensor, developed by Clark and Lyons in 1962 [5]. After that, the first commercial blood glucose analyzer was released by Yellow Springs Instruments (YSI) in 1975 and the first glucose meter by Medisense years later, in 1987. Since then, the interest in developing glucose biosensors and their impact in the biosensor market have grown enormously (glucose sensors or glucose meters represent an 85% share of the biosensor market) [6] and continue to rise (Figure 1), being one of the point-of-care devices par excellence in the biosensor world [7]. This increasing interest in achieving even smaller and cheaper glucose sensors, keeping or even improving their characteristics, such as long-term stability, specificity and biocompatibility, among others, promotes the implementation of the latest sensing trends and technologies in their development (e.g., 3D printing or wearable approaches) [8,9,10].

The most popular glucose biosensors use enzymes as biorecognition elements, and electrochemical techniques for detection. Although, in most of the cases, more than one enzyme is needed, this strategy is very simple and highly specific and sensitive (because of the enzymatic amplification produced). Regarding transduction, amperometry is the electrochemical technique most widely used in glucose electrochemical biosensors and it is characterized by its small energy consumption, ease of use and simplicity (just applying a constant potential for a time is required). [13]. In terms of material, one of the strategies that has been used to develop simpler and cheaper glucose sensors is the use of paper, with a dramatic increase in the last ten years (Figure 1a). This is due to its numerous advantages, apart from low cost: its presentation in light sheets, which makes easy its transportation and storage; its foldability that allows multilayer/origami designs; its porosity that enables bio/nanomodifications, as well as fluid transportation; and its biodegradability, which converts it into an environmentally friendly material. In this review, a comprehensive literature review is carried out, starting with the use of paper as substrate for biosensing, considering the different types of enzymes that have been employed, and including the most interesting paper-based designs and characteristics of glucose biosensors found in the literature in recent years.

## 2. Paper as a Substrate for Biosensing

Paper is a widespread material, essential for communication and for documenting our past. Therefore, paper fabrication is one of the greatest landmarks in human history, whose origin is traced back to China in the 2nd century AD. In addition to this, paper has been employed as substrate for analytical tests for centuries [14]. The first paper-based chemical sensor was probably the litmus paper for pH measurement, assumed to be derived from the essays, dated to 1664, of Robert Boyle, which related the color change of flower and vegetable dyes with the acidity of solutions [15]. However, the pH test strips were not patented and commercialized until the 1920s [16]. Before this, in the 1860s, the first urine test was reported for monitoring glucose levels of diabetes patients [17]. After these early reports, since 1940s, paper has become increasingly used in Analytical Chemistry for chromatography applications [18,19]. In 1956, the first latex agglutination was developed by Plotz and Singer, which is the basis of the lateral flow immunoassay (LFA) for the diagnosis of rheumatoid arthritis [20]. From that date, the basic principles of lateral flow technology were refined and, in the latter years of the 1980s, companies such as Unilever and Carter Wallace filed several important patents [21,22]. Currently, the simplicity of LFAs has increased enormously the interest in their use, as demonstrated by the well-known pregnancy test and the now widely used COVID-19 diagnosis test. Despite being a material employed for analytical tests long ago, paper has been rediscovered in recent years as substrate material for developing analytical devices since, in 2007, Whitesides et al. introduced the concept of Microfluidic Paper-based Analytical Devices (µPADs) [23]. In that work, paper was patterned by photolithography, defining hydrophilic flow channels to perform multiplexed assays. Although it was not strictly a new idea, since, in 1937, Yagoda reported the first filter paper with a hydrophobic pattern, using paraffin to delimit the detection area [24], and, in 1949, Clegg and Müller also used paraffin to design a fluidic channel in filter paper with chromatographic purposes [25], it was Whitesides’ work that opened up a large new area in Analytical Chemistry. From that moment, the development of paper-based devices has experienced a huge growth [16,17,26,27,28]. The intrinsic properties of paper (e.g., biocompatibility, porous nature, flatness, and presence of capillary forces) make new advances possible, not only in clinical and diagnostic fields, but also in food and environmental applications. Apart from these features, the success of paper-based analytical devices is mainly associated with their low cost, simplicity of fabrication and operation, portability, and disposability. The great versatility, demonstrated in lateral and vertical flow devices, single or multi-layered devices, which can be combined with other flat substrates (e.g., tape), together with the possibilities of being cut, pressed, folded, or printed or serving as excellent storage for biocomponents, conductive elements or nanomaterials, makes this material extremely appropriate for the decentralization of analysis. Hence, only in the period from 2016 to 2021, around 22200 scientific references including the term “paper-based” have been reported (Scopus database).

### 2.1. Fabrication and Characteristics of Paper

Paper is usually fabricated using cellulose fibers, but it can also be fabricated using non-cellulosic porous materials, such as glass fiber or polyester. Most paper is fabricated by a wet process to release the cellulose fibers from the wood through a mechanical and/or chemical process, and to mix them with water to a concentration of ca. 0.1–1% [29]. Then, water is removed, keeping the fibers distributed in such a way that a homogeneous paper sheet is obtained. The final composition of paper varies, depending on the type of pulping process. When mechanical pulping is performed, the final paper contains cellulose, lignin and pitch (e.g., as in newsprint paper), while paper obtained by chemical pulping contains residual lignin and has a brown color (e.g., as in grocery bags) [30]. The papers employed as office paper and as filter paper are obtained from fully bleached pulp.

The major component of most paper types is cellulose, which is the most abundant biopolymer and consists of a linear chain of hundreds of glucose units [30,31]. Cellulose is fibrous, tough, insoluble in water and biodegradable. In paper, cellulose fibers (hollow tubes about 1.5-mm long, 20-μm wide, with a wall thickness of 2 μm) are layered in the *x*,*y* plane after the filtration process, with an orientation that depends on the paper-making process. Then, fluid transport in a paper strip depends on the angle at which the paper is cut [30].

Paper is defined by two main properties correlated between them: the thickness and the basis weight (mass of dried paper per square meter). Thus, the density of paper can be estimated from these two properties. The geometry and the hydrophilic nature of paper determine its main fluidic properties [17,27,29] that are of critical importance in paper-based bioanalysis: e.g., the immobilization of bioreagents is influenced by the porosity and the accessible surface area (Table 1). Moreover, different additives that can be used to modify the final characteristics of the paper [32] can influence the results. For example, brightening agents (usually fluorescent) used to improve the whiteness of paper may provide high background signals in fluorescence-detection assays, while polymers to enhance the wet-strength of paper are interesting for biological reactions, since most of them are performed in liquid phase.

### 2.2. Main Paper Features in Relation to Bioanalysis

Worldwide availability and low cost make paper a useful substrate for developing disposable and portable point-of-care diagnostic devices [33]. It can be mass-produced, allowing easy adaptation to a large-scale production of analytical devices. While it lacks the high mechanical robustness of traditional microfluidic materials (e.g., glass or polymers), it has many more advantages [14]. It is thin and light, making its storage and transport easy. It is flexible and compatible with many printing technologies making it very useful for µPAD fabrication. Its flexibility also allows stacking and folding to create multi-layered or origami 3D structures. Moreover, paper is usually white, which is very useful for colorimetric assays since it provides a strong contrast with the colored product of the assay. Paper can be acquired in a wide range of thicknesses and engineered forms, allowing its use for different purposes: paper with a well-defined pore size can be used as filter, or paper modified with conductive fibers or inks (carbonous or metallic) can conduct electricity [34]. Furthermore, it can be easily modified with functional groups or nanomaterials and, since it is made of cellulose, it is compatible with biological samples and allows the immobilization and storage of bioreagents in active form. In addition to all this, paper is biodegradable and, therefore, eco-friendly. Being flammable, it is useful for bio-contaminated samples, since it permits easy and safe disposal by incineration.

As was commented before, particular attention must be paid to the use of additives, as well as to the possible variation in composition between different sheets, even from the same supplier. Its high absorption capacity, which allows its easy modification, makes residue removal difficult, and makes it highly sensitive to moisture. This can change its dimensions and, therefore, its spatial resolution and mechanical strength [35]. This hindrance can be overcome by treating paper to make it hydrophobic [36]. It has also to be kept in mind that its combustion temperature around 200 °C, which makes it incompatible with some common microfabrication techniques. Therefore, several factors must be addressed when developing paper-based biosensors, such as fluidic design, transducer signal generation or surface immobilization chemistry, among others [37], to take advantage of the great potential they possess to achieve point-of-care (POC) analysis. These will play a relevant role not only in developed, but also in developing countries for which the World Health Organization indicates that diagnostic devices should be REASSURED, i.e., provide real-time connectivity and ease of specimen collection, in addition to being affordable, sensitive, specific, user-friendly, rapid and robust, equipment-free and deliverable to end-users [38].

In more detail, electrochemical paper-based analytical devices, those where detection is based on the use of electrodes for measuring interfacial electrical properties, are highly connected with all these characteristics. The instrumentation required is easy to miniaturize, cheap and very simple, which makes it portable and autonomous. Therefore, the combination of the great advantages of paper as substrate with the sensitivity and selectivity of electrochemical detection has become a highly exploited research field. The properties and characteristics of paper allow it to play different roles in electroanalytical methodologies, mainly as substrate for printing electrodes of the electrochemical cell, but also for filtration, separation, or storage of reagents, approaching the concept of lab-on-a-chip devices, i.e., lab-on-paper in this case. Figure 1c represents in a schematic way the main roles paper plays in electrochemical paper-based devices, showing the wide functionalities of paper.

### 2.3. Types of Paper Used in Electroanalytical Devices

Currently, a great variety of paper materials can be found, being cellulose fiber-based materials and nitrocellulose membranes the most employed in paper-based analytical devices [39]. Regarding cellulose fiber-based papers (cellulosic paper), such as filter and chromatography paper, Whatman^®^ filter papers are particularly popular. Among them, the Grade No. 1, which offers medium retention and flow rate, is the most employed [14,28,36,40,41,42,43,44,45] (Table 1).

Nitrocellulose membranes, well-known as the key substrate for LFAs, have also been used for developing electroanalytical platforms for years [46,47,48]. This material is produced by partial nitration of cellulose, and in contrast to cellulosic papers, nitrocellulose membranes are hydrophobic. Although they offer a stable and reproducible liquid flow, the wax penetration used to define hydrophobic channels is slow compared to filter paper [49,50,51]. However, this can be overcome by cutting paper in one-dimensional strips, since in this way flow direction does not need to be delimited with wax. 

In addition to these paper materials, other types, such as glossy paper (hydrophobic, resistant to chemical attack and degradation) [52], paper towel (high absorptivity and poor mechanical stability) [53,54] and office paper (cheaper and more available) [55], have been employed in sensor technologies [56]. For example, different types of paper have been compared by Martín-Yerga et al. [57] for developing hand-made screen-printed electrodes, and by Nunez-Bajo et al. for manufacturing gold-sputtered electrofluidic paper [58]. The use of chemically modified paper as substrate is also of great interest because of the different characteristics it can offer. For instance, ion-exchange cellulose [57,59] and polyester–cellulose blend [54,60,61] papers are commercially available [39]. Moreover, the great versatility of paper allows other modifications to achieve the required physical characteristics. In this context, Whatman^®^ paper can be silanized with fluoroalkyltrichlorosilanes, obtaining a hydrophobic and oleophobic (i.e., omniphobic) substrate that preserves its mechanical flexibility [32].

**Table 1 sensors-22-06232-t001:** Features of different types of paper employed as substrate for paper-based (electro)analytical devices. Reprinted from [58], Copyright (2017), with permission from Elsevier.

Paper	Material	Thickness (µm)	Volume/Area (µL/cm^2^)	Flow (mm/min)
Whatman Grade 1Chr	Cellulose	180	9	4.3
Whatman Grade 3MM Chr	Cellulose	340	15	4.3
Whatman Grade P81	Cellulose with phosphate groups	200	7	3.2
Whatman Grade DE81	Cellulose with diethylaminoethyl groups	230	7	4.2
Millipore Hiflow plus	Nitrocellulose	131	11	13.3
Mdi Type PT-R5	Polyester	325–485	25	66.7
Mdi Type GFB-R7L	Glass fiber	520–680	25	28.6
Whatman GF/F	Glass fiber	420	21	

In addition to these materials, in recent years, and with the advances in nanomaterials, cellulose nanomaterials, or nanocelluloses, have attracted huge attention for electroanalytical applications [56,62]. Mainly, they can be classified into three main types, depending on the source of the cellulosic material, preparation methods, processing conditions, morphological features and functions: cellulose nanocrystals, nanofibrils and bacterial nanocellulose [62,63,64]. They offer outstanding properties, such as excellent chemical-modification abilities, extraordinary mechanical strength, and high thermal stability, among others. Moreover, they are available in a wide variety of morphologies (nanofibrils, nanocrystals, nanofilms and electrospun cellulose fibers) and forms (transparent films, hydrogels, aerogels, spherical particles, etc.) [62].

Other cellulose-based materials that have been employed to construct analytical devices are cotton cloth, yarn [65,66] and lignocellulose [67]. Vegetal parchment has been used as a substrate for screen-printing carbon electrodes [68], while coated and uncoated cellophane (a polymer derived from cellulose) has also been used for the fabrication of open-channel microfluidic devices [69], printing electrodes on coated cellophane, and developing resistive heaters and different electroanalytical devices for their use in flow injection analysis and electrochemiluminescence applications.

## 3. Enzymes

Enzymes are biological catalysts able to accelerate any chemical reaction, not being consumed in the process nor being part of the final products. Enzymes decrease the activation energy of reactions, showing a great specificity, in contrast to what happens with inorganic catalysts. Enzymes only act on one specific substance (or group of substances), also called the substrate. Their high specificity makes them capable of differentiating even between different stereoisomers of one compound; for instance, between D- or L-glucose [70].

### 3.1. Types

Their name is usually built by adding “-ase” to the name of the substrate whose reaction they catalyze. They can also be named by defining the type of reaction, being this principle the most logical one for the classification of enzymes. In order to unify all the types of enzymes, the International Union of Biochemistry and Molecular Biology published a general classification in which six main groups (European Comission, EC numbers) were distinguished [11] (Figure 1b): oxidoreductases (EC1), which catalyze redox reactions, and include oxidases, dehydrogenases, peroxidases and oxygenases; transferases (EC2), which catalyze the transfer of functional groups (amino, carboxyl, methyl, etc); hydrolases (EC3), which catalyze the cleavage of bonds such as C–O, C–S, C–N or O–P; lyases (EC4), catalyzing the cleavage of bonds such as C–C, C–S or C–N, excluding peptide bonds; isomerases (EC5), that interconvert any type of isomer, both optical, geometric or positional; and ligases (EC6), which catalyze the binding of two different molecules, performing hydrolysis of a high-energy bond of nucleoside triphosphate, forming bonds such as C–C, C–S, C–O or C–N [71]. In 2018, a seventh group of enzymes was discovered, known as translocases, which catalyze transfers from trans- to cis-conformation, or vice versa [71]. This type has also been added to the general classification.

Other types of catalysts that, in recent years, have attracted great attention are pseudoenzymes, which are proteins with a sequence homologous to that of enzymes, but without enzymatic activity, due to mutations in catalytic amino acid residues. Although they do not show catalytic activity, they can be found in many metabolic and signaling reactions combined with other enzymes, by different mechanisms. Regulating catalytic yields of conventional enzymes or binding substrates to assure the reaction between them and the enzymes are some examples. The most well-known are pseudokinases, although other such as pseudophosphatases, pseudolyases or pseudoproteases can be found [72].

Natural enzymes are widely used in many fields, but, sometimes, they present low stability, poor biological variety, and high costs. Due to this, a great effort to find new artificial reagents which mimic the activity of enzymes has been made. An example is nanozymes, which are usually made of nanomaterials with catalytic properties, and combine the advantages of typical chemical catalysts and biocatalysts. Although this review is focused on enzymatic glucose biosensors, it is important to know that there is increasing research on this type of nanomaterials, whose use is spreading in biosensing, and which are possible alternatives to enzymes because of their low cost and high stability [73].

### 3.2. Immobilization Procedures

A critical step for designing enzymatic electrochemical biosensors is the immobilization of the enzymes on the surface of the transducer because the chosen immobilization method depends on the required characteristics of the sensor, not only the analytical ones (e.g., sensitivity, precision, stability), but also others such as simplicity and the final cost of the device. Different immobilization methods used to develop enzymatic biosensors [74,75,76] are commented on in the following sections.

#### 3.2.1. Adsorption

Adsorption is the simplest method of immobilization. It is generally carried out by dissolving the enzymes in a buffer solution and putting it in contact with the solid support for a period of time. The bonds formed in this immobilization approach are weak, such as Van der Waals’ forces, and electrostatic or hydrophobic interactions. The main drawback of this strategy is that any change in the conditions of the medium, such as pH, temperature, or ionic strength, can produce enzyme desorption, affecting the stability of the biosensor. Adsorption can be carried out in three different ways: (i) physical adsorption, when the enzyme solution is directly deposited on the surface of the electrode; (ii) electrostatic interactions, when the enzymes are immobilized onto a charged surface; and (iii) retention in a lipidic microenvironment, using Langmuir–Blodgett (LB) technology, with accurately controlled thickness and molecular organization, where enzymes can be easily adsorbed [77].

#### 3.2.2. Entrapment

Immobilization of enzymes can be carried out inside a three-dimensional matrix, by an easy methodology in which enzymes, mediators and additives can be simultaneously deposited on the surface of the electrode. The activity of the enzymes is preserved since there is no modification of the biological elements. The entrapment can be carried out by different methods, but the most used is electrochemical polymerization. It consists of applying an appropriate current or potential on an electrode with a mixture of the enzyme and monomer, in a way that the enzyme is physically incorporated within the polymer network. The most widely used polymers are conducting polymers, such as polyaniline, polypyrrole or polythiophene. Enzymes can also be entrapped in polysaccharide-based gels, carbon paste or sol-gel matrixes, forming solid metal or semimetal oxides by using hydrolytically labile precursors, such as polysilicic acid, halides of aluminium or polysilicic acid.

#### 3.2.3. Cross-Linking

Another useful approach for the immobilization of enzymes is cross-linking, using reagents such as glutaraldehyde or other bifunctional agents, such as glyoxal or hexamethylenediamine. It is a very simple method which enables a strong chemical binding. Other cross-linking agents, such as functionally inert proteins (e.g., bovine serum albumin) can also be used.

#### 3.2.4. Embedding and Encapsulation

The encapsulation of enzymes in liposomes is another approach used in enzymatic sensors. This strategy allows the stabilization of the enzyme, maintaining its activity for longer times and, consequently, improving the performance and the stability of the sensor [78,79]. Enzymes can also be embedded in metal–organic frameworks (MOFs), which have recently gained a lot of popularity as porous hybrid materials, because of their great stability, low density, crystallographic structure and high pore volume. Apart from that, it has shown to have great potential as an enzyme support due to its high surface area and the strong interactions between the organic part and the enzymes, preventing them from leaking. The design of MOFs for immobilizing enzymes can follow two different strategies, depending on the need of having co-precipitation agents. On the one hand, the co-precipitation method requires stabilizers to ensure that the enzyme is in its active form during the preparation. In this case, the enzyme is encapsulated with a co-precipitating agent (such as polyvinyl pyrrolidone, PVP). On the other hand, biomimetic mineralization can be carried out in the absence of co-precipitating agents to form enzyme–MOF biocomposites [80].

#### 3.2.5. Affinity Reactions

Oriented immobilization of enzymes can be carried out creating affinity bonds between activated supports and specific groups of the peptide sequence of the enzyme. A great amount of affinity approaches has been described to immobilize enzymes, being avidin–biotin, lectin–carbohydrates or metal cation–chelator interactions the most used. The enzyme can contain affinity tags naturally within its sequence, or it can be inserted prior using chemical or genetic engineering methods.

#### 3.2.6. Covalent Immobilization

The covalent coupling between enzymes and polymeric substrates is very popular in the development of enzymatic biosensors. It consists of the binding of the enzymes to the surface by means of functional groups which are not involved in the catalytic activity. In order to perform a covalent immobilization, a previous activation of the surface is usually carried out using multifunctional reagents, such as glutaraldehyde or carbodiimide. The enzyme reacts with the activated support, binding covalently, either to the transducer surface or onto an activated membrane made of synthetic polymers, such as Nylon^®^, or natural matrixes, such as cellulose.

### 3.3. Enzymatic Sensors

Among the different possibilities within enzymatic assays (for saccharide determination), different “generations” of enzymatic sensors have been described, depending on how enzymes are used [81] (Figure 2).

First generation: their principle is based on the use of oxygen as a co-substrate, producing H_2_O_2_ as the enzymatic reaction product, which is measured. However, the electrochemical monitoring of H_2_O_2_ requires high operating potentials, which can affect the selectivity, due to the high probability of interferences, i.e., electroactive species that can be reduced at those high potentials.Second generation: they overcome the limitation coming from the high potential needed for the determination of the H_2_O_2_ by using redox mediators as co-substrates, which ensure the efficient electron transfer at lower potentials and are regenerated on the surface of the electrode. Mediators are artificial electron transfer agents which can participate in the reaction with the enzyme and increase the electron transfer rate. An ideal mediator should: (i) be able to react quickly with the enzyme; (ii) have reversible kinetics; (iii) be pH independent; (iv) have stable reduced and oxidized forms and (v) do not react with oxygen in any of its forms [82]. Different electron mediators have been used, being ferrocene, ferrocyanide, cobalt phtalocyanine and methylene blue among the most common.Third generation: they do not need mediators since they are based on the direct transfer between the enzyme and the surface of the electrodes at very low operating potentials, achieving high selectivity.Fourth generation: they are enzyme-free sensors in which the glucose is directly oxidized on the electrode surface [83]. Although numerous enzyme-free sensors for glucose have been reported, mainly using electrocatalytic nanostructures based on transition elements, the performance of enzymatic glucose sensors is still better in terms of sensitivity and biocompatibility.

**Figure 2 sensors-22-06232-f002:**
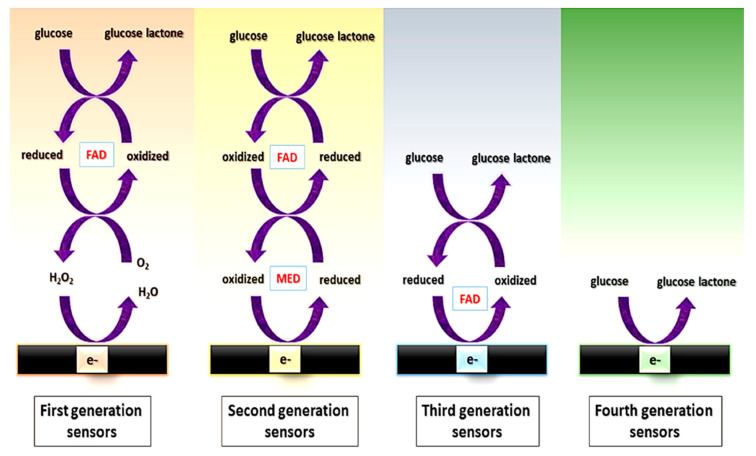
Scheme of the electron transfer pathway between the enzyme and the electrochemical cell for first, second, third, and fourth generation of glucose sensors. FAD: Flavin group. MED: electron transfer mediator. Reprinted from [83] (open access).

Among all the enzymes that can be used for glucose determination, the most common is glucose oxidase (GOx), followed by glucose dehydrogenase (GDH) [84,85], both belonging to the EC1 (oxidoreductases) group. GOx is a dimeric enzyme with flavin adenine dinucleotide (FAD) as its redox prosthetic group. GOX oxidizes glucose to gluconic acid using oxygen as the electron acceptor and producing H_2_O_2_ [86]. This reaction involves the reduction of the flavin group to give the reduced form of the enzyme (FADH_2_). GDH is an oxidoreductase enzyme which also oxidizes glucose into gluconic acid, but it is unable to use oxygen as an electron acceptor; therefore, it requires the cofactors (e.g., nicotine adenine dinucleotide (NAD), nicotine adenine dinucleotide phosphate (NADP) or pyrroloquinoline quinone (PQQ)) [84,85]. Since the potential needed to detect the products of these enzymatic reactions is high (e.g., +0.6 V for H_2_O_2_), it is common to use auxiliary reactions forming a cascade of enzymatic reactions that enhance the sensitivity of the sensor. The most commonly used enzymes for this purpose are peroxidases which catalyze the decomposition of H_2_O_2_. Moreover, the use of redox mediators, such as ferrocyanide [87], Prussian Blue or cobalt phthalocyanine [88], is also common to decrease even more the overpotential required, with the aim of minimizing the oxidation of endogenous species [85].

## 4. Transduction Step

Colorimetric detection is very appropriate for white paper-based analytical devices. However, this type of detection is not usually sensitive enough, and it depends on many factors, such as, for example, the environmental light. By contrast, electrochemical detection is an alternative that provides higher sensitivity and does not depend on illumination or sample color. In addition to this, electrochemical techniques show several advantages for µPADs: (i) electrodes can be easily miniaturized and manufactured, (ii) the instrumentation needed is not very complex nor expensive, (iii) portable potentiostats are nowadays commercially available, and (iv) electrochemical detection can be applied in many different fields, such as food, medical or environmental [89]. The excellent integration between paper as substrate and electrochemistry for detection has stimulated the development of electrochemical paper-based analytical devices (ePADs), with a huge growth in the last 10 years.

In this context, the first ePAD, reported by Henry’s group, used photolithography to design hydrophobic channels and screen printing to fabricate the electrodes [90]. In the development of ePADs, apart from the design and the patterning, electrode fabrication is a critical step, since the electron transfer is an interfacial phenomenon, and the electrode surface determines the analytical characteristics of the device. An electrochemical transducer usually consists of a cell with two electrodes (reference and indicator, mainly used for potentiometric measurements) or three electrodes (reference (RE), counter (CE) and working (WE) electrodes, mainly for voltammetric techniques). The use of paper, with extremely interesting properties, allows many design possibilities.

The electrochemical techniques that are used for detection in electrochemical biosensors include a wide range of measurement possibilities [91]. Thus, electrodic techniques, those based on the measurement of electrical properties at the electrolyte–electrode interface, allow either static (current-zero measurements as in potentiometry) or dynamic (with current flowing through the system as in amperometry) approaches. The most common are amperometric techniques, where the current is measured at a given potential (named amperometry or chronoamperometry, depending on the mass transport regime) or during a potential scan (in this case, they are known as voltametric techniques). In the last case, a linear scan can be made (cyclic voltammetry or linear sweep voltammetry) or pulses can be superimposed, as in the two most common modes: differential pulse voltammetry (DPV) and square wave voltammetry (SWV). The current is sampled before and after the pulse in DPV or before the end of forward/backward pulses in SWV, which decreases the capacitive current, and, in turn, increases the sensitivity (proportional to the i_f_/i_c_ ratio). In those cases, the measurement is made under a diffusional mass transport regime, but convection is possible (hydrodynamic voltammetry). On the other hand, amperometry can be performed under diffusional or convective regimes. In the first case, chronoamperograms (i-t curves) are recorded and the current is usually measured at the stationary and more precise part of the curve (e.g., 20 s), where capacitive current is almost negligible. Injection analysis either under flow (FIA) or static systems (BIA, batch injection analysis) produces transitory signals, i.e., fiagrams or biagrams. The measurement of the current requires the use of a potentiostat to apply the excitation signal to the working electrode (usually included in a potentiostatic system of three electrodes, together with RE and CE) and to record the response signal (i-E or i-t curve). Regarding potentiometric techniques, as mentioned before, they require a two-electrode system to measure the difference of potential with a potentiometer.

The transducer and its design are key components of an electrochemical biosensor. In this section, different designs of transducers are briefly summarized, distinguishing between 2D and 3D paper-based devices, and describing different approaches to include or integrate electrodes on paper. Most of these approaches take advantage of the porous nature of paper, modifying it with a conductive material, usually in the form of ink (made of a carbon or metallic material), through different techniques such as screen/stencil printing, inkjet printing, drawing, or sputtering. Other strategies combining metallic (micro)wires and paper have also been reported [41,87,92,93]. The incorporation of nanomaterials for improving the performance of ePADs is a very common strategy since these materials can enlarge the electroactive surface and enhance the electron transfer and the immobilization of biomolecules. The modification of paper-based carbon electrodes with nanomaterials such as nanoparticles, including metallic (e.g., gold, silver, platinum) and metallic oxide (e.g., zinc or nickel oxide), to improve their electrochemical characteristics has been widely reported. Carbon nanomaterials are usually used as transduction materials on ePADs because of their excellent conductivity, but also due to their high surface area, which boosts the anchoring of biomolecules. The high porosity and tunable structures of MOFs have made them attractive for developing ePADs. An example is the origami ePAD based on a cobalt metal–organic framework of modified carbon cloth/paper developed by Wei et al., for non-enzymatic glucose determination, in which the Co–MOFs act as an oxidase-mimicking nanozyme [94]. Several interesting articles have reviewed the different reported strategies to fabricate electrodes on paper [95,96,97] and the main usages of nanomaterials on ePADs [98,99,100].

In this section, as well as reviewing different approaches for developing ePADs, the possibility of using paper as a storage support is also considered. 

### 4.1. Patterning Paper-Based Devices

When talking about paper-based microfluidics, fluid handling takes place by passive wicking or pumping, due to the high capillarity of paper [101]. In two-dimensional paper-based microfluidic devices, patterning is based on the change from hydrophilic to hydrophobic, in order to make one-dimensional channels, where the liquid flows due to capillary action [102].

One of the most widely used strategies for creating hydrophobic barriers to delimit the passage of fluids only through the area designated with such purpose is wax printing [103]. In paper devices created using this type of patterning, power supplies or external propulsion systems are not usually needed [104,105]. The procedure for wax printing is very simple: in the first step, wax ink is deposited on the paper, usually employing a wax printer; then, it is necessary to heat the paper to assure the diffusion of the wax within the paper fibers, creating the hydrophobic barriers. Wax patterning can be also three-dimensional, using open-channels, hemi-channels or fully enclosed channels, depending on how the wax is melted [106]. Another similar strategy is inkjet printing, in which an inkjet printer is used to transfer specific reagents such as alkyl or alkenyl polymers to the paper [107,108].

As alternatives, there are several works in which authors use photolithography, the methodology used in the first μPAD, described by Whitesides’ group [23]. In order to perform a low-cost methodology, photolithography was used by Yu and Shi [109] to create microchannels on Parafilm^®^, embossing them onto paper. Nevertheless, this method usually requires expensive equipment and organic solvents which can affect the characteristics of the paper. Polymers (e.g., polyurethane acrylate [110] or polystyrene [111]) have also been used in order to make paper hydrophobic, and open channels have been set out by changing the surface wettability by patterning channels with different geometries and chemical compositions [112]. Different approaches have been developed, such as plasma treatment or fluoro-silanization [113], among others. Commercially available permanent markers have been also used to design hydrophobic channels [58,114].

On the other hand, physical patterning can also be carried out. In this context, a knife plotter can be used for precisely cutting the paper [115], or a laser cutting (e.g., CO_2_ laser cutter [116,117]) to obtain very thin microfluidic patterns when using optimized parameters. There are also examples where patterns or stamps are not required because one-dimensional cellulosic materials such as thread, combined with paper, are used [118,119,120].

### 4.2. 3D Paper-Based Analytical Devices

The techniques for patterning paper previously described are widely applied to construct 2D µPADs or static devices, in which the solution is put onto paper-based electrodes for recording the measurement. On the other hand, dynamic µPADs involve the transport or addition of reagents, the creation of integrated reservoirs for sample or waste, and/or the combination of different sensing phases for different analytes. The more recent trend to incorporate several steps of the analytical process is the design of 3D µPADs, which can be done following different approaches [121,122]:Layer stacking or multilayer: It consists of piling up different layers of patterned paper, following some of the strategies described before. The flow direction can be both horizontal, as in lateral flow assays, or vertical, as in the paper-based optical glucose biosensor developed by Chun et al. [123]. They formed hydrophobic barriers by wax printing and then made a multilayer platform with different loading and detection zones. Another example is the paper-based microfluidic device for glucose monitoring that uses MOFs to immobilize the enzymes and a 3D µPAD placing paper-based wax-printed chips for detection between polyvinylchloride layers [124]. Layers of materials different from paper could be included; for example, adhesive tape and hollow channels [125] could be added to improve fluidics.Origami: This is a special multilayer approach where the layers are connected by folding [126]. This approach has been applied in many paper-based devices, taking advantage of one of the inherent characteristics of paper: flexibility. Examples of this approach are the 3D paper devices combining wax printing and origami techniques for the detection of protein and glucose in urine designed by Sechi et al. [127], or the one developed by Xie et al. using polydimethylsiloxane (PDMS)-coated folded paper [128]. In this case, PDMS serves as the patterning reagent that is cured in the paper. Other works describe the use of folding paper in order to obtain electrophoretic separations [129,130].3D printing: although the most common strategies to design three-dimensional paper-based devices are stacking or folding, 3D printing has been also used for the construction of 3D µPADs, placing the paper between two layers of 3D-printed designs. Fu et al. [131] studied how controllable the flow is when these microfluidic analytical devices are fabricated by 3D printing technology, developing a colorimetric assay for glucose and albumin with limits of detection low enough for clinical purposes. A 3D-printed holder was also used by de Castro and coworkers [132] for the development of a wearable sensor for glucose and nitrite monitoring.

### 4.3. Fabrication of Electrodes on Paper

There is a great variety of methods reported for the fabrication of electrodes on paper platforms [12]. Regarding the paper-based enzymatic biosensors for the determination of glucose, in most cases, film-based electrodes are employed. Then, a thick-film methodology (µm-thick layers) could be employed, commonly associated with the use of inks. However, thin-film methodologies, to obtain nm-thick layers, not as common as the use of inks, are also possible, as in, e.g., gold-sputtered electrodes [58]. Apart from the use of film-based electrodes, wire electrodes have also been combined with paper-based electrochemical cells [133], although the use of printing technologies remains the most intuitive and common.

#### 4.3.1. Screen or Stencil Printing

Screen printing was the technique used for the first ePAD developed in 2009 [90] (Figure 3a), and it is the most common strategy employed for incorporating electrodes on paper. A screen or stencil, which is usually made of polymers, such as Nylon^®^, with the desired electrode pattern is required. The conductive ink is forced to pass through this stencil, by pressing with a squeegee or using specific machines, transferring the pattern to the substrate. In the first reported ePAD that has already been commented on, a first layer with silver/silver chloride ink (for connections and reference electrode) was deposited and then, carbon ink containing the mediator Prussian Blue was applied on working and auxiliary electrodes. As a proof of concept, this design was used for the simultaneous determination of glucose, lactate and uric acid using oxidase enzymes (glucose oxidase, lactate oxidase and uricase, respectively). In the case of glucose, the analyte of interest in this review, the linear dynamic range was up to 100 mM and the limit of detection (LOD), 0.21 mM. Since then, screen-printing technology has been used for developing ePADs for years. In 2013, the same analytes (glucose, lactate and uric acid) were determined by Zhao et al. [134] (Figure 3b) using a microfluidic paper-based electrochemical array, patterned by wax printing and using a handheld custom-made potentiostat for signal readout. This device provided a dynamic linear range for glucose comprised between 5 and 20 mM, with an LOD of 0.35 mM.

Within the aim of developing micro-total analysis systems (µTAS), integrating different operations in the paper-based device, Cinti et al. developed, in 2018, a glucose electrochemical biosensor for whole blood analysis [135] (Figure 3c), which used filter paper in which Prussian Blue nanoparticles were synthesized using very low volumes of precursors and with no need of external inputs. In this modified paper, whose authors called “Paper Blue”, wax and screen-printing technologies were combined for manufacturing the glucose biosensor using glucose oxidase as the recognition element. Glucose was detected amperometrically at concentrations up to 25 mM, with an LOD of 0.17 mM and a sensitivity of 1.70 µA·mM^−1^. This biosensor was applied to the determination of glucose in real blood samples, monitoring normal or pathological conditions related to glucose levels in whole blood. The µTAS approach was also exploited [136] for the development of an ePAD for the detection of glucose that contained different reservoirs for the buffer, electrochemical mediator (ferricyanide) and enzyme (FAD-dependent glucose dehydrogenase, in this case), which were successively mixed after inserting the sample by dropping. They obtained a wide linear range, between 0.5 and 50 mM, with an LOD of 0.33 mM.

Multiplexed analysis is also becoming very important, since it allows us to make simultaneous measurements, saving time and decreasing costs considerably. A very interesting example is the microfluidic paper-based multilayer device based on sixteen independent channels with electrochemical detection proposed by Fava et al. [137]. The performance of this device was characterized electrochemically using ferrocenecarboxilic acid as an electroactive probe. Then, with this device, an enzymatic sensor for the determination of glucose in urine samples was developed using glucose oxidase as the enzyme. The biosensor obtained was able to measure glucose in a concentration range from 0.1 to 40 mM, obtaining an LOD of 30 µM. Cao et al. [138] developed another multilayer device with two layers, combining photolithography and screen-printing techniques: in the first layer, where the auxiliary and the reference electrodes were located, glucose oxidase was immobilized; whereas, in the second one, containing the working electrode, highly conductive Prussian Blue deposited on reduced graphene oxide–tetraethylene pentaamine (rGO–TEPA/PB) was used to modify the electrode and as an electrochemical-sensitive membrane for the determination of hydrogen peroxide. The calibration range obtained with this biosensor was between 0.1 and 25 mM, with a detection limit of 25 µM.

As commented before, three-dimensional electrochemical paper-based analytical devices have gained importance in recent years. This is achieved not only with layer stacking, but also with origami approaches. For example, Punjiya et al. [139] (Figure 4a) presented a three-dimensional paper-based device using a hollow 3D fluid reservoir which transports the sample to the electroactive area. Here, the techniques of wax printing, screen printing and paper folding were combined, and the obtained device was connected to a custom-designed potentiostat as a miniaturized reader. This sensor was used for the determination of important biochemical analytes, such as dopamine or glucose, as well as pH. In the case of the chronoamperometric detection of glucose, the enzyme glucose oxidase was immobilized by drop casting and glucose was dissolved in a mediator solution. The methodology showed linearity within the range of 5 and 17.5 mM, and a sensitivity of 0.34 µA·mM^−1^. An interesting wearable origami-based device for sweat analysis was fabricated by wax and screen printing on paper folded to create five different layers: a sweat collector, vertical channel, transverse channel, electrode layer and sweat evaporator [140] (Figure 4b). This wearable glucose sensor demonstrated the capability of collecting, analyzing and evaporating sweat, thanks to the capillary action of the paper and the hydrophobicity of wax. The methodology showed a linear response between 0.25 and 1.5 mM, and an LOD as low as 5 µM. It was proved with on-body measurements of glucose in sweat.

A combination of a folding-based device and a multilayer platform was approached by Wu et al. [141], using layers of adhesive films and folding paper layers, permitting different assays in several layers. The 3D paper analytical device developed here allowed both colorimetric and electrochemical detection for the quantification of glucose in a range from 1 to 40 mM, with an LOD of 0.32 mM, lower than that obtained with the colorimetric detection (1.35 mM). After demonstrating the ability of this design for developing biosensors, an automated and multiplexed ELISA for the detection of troponin I was carried out, increasing the number of layers to include the washing steps.

Pesaran et al. designed a 3D origami design of a paper-based potentiometric sensor, in which carbon paste was put on the paper by pressing it through an iron mold and a magnet. By modifying the ePAD with GOx, it responded to glucose concentrations in a range from 1 nM to 0.1 mM [142].

#### 4.3.2. Inkjet Printing

Commercial inkjet printers can also be used for the construction of transducers of electrochemical biosensors. Although they have some advantages, such as their high availability and the simplicity of the method, there are not many examples of inkjet-printed electrodes for the detection of glucose. One of the first was reported by Määttänen et al. [143], who printed nanoparticle-based gold working and counter electrodes, and an electrogenerated layer of Ag/AgCl as a quasi-reference electrode. The three-electrode area was then defined applying a hydrophobic ink made of polydimethylsiloxane (PDMS). After characterizing the electrodes by cyclic voltammetry of the ferrocyanide/ferricyanide redox system, this device was used to develop a glucose biosensor using polyaniline (PANI) as the conducting polymer and glucose oxidase entrapped in poly-3,4-ethylenedioxythiophene (PEDOT) films. The current response showed linearity up to a glucose concentration of 5 mM.

#### 4.3.3. Pencil Drawing

One of the easiest and simplest ways of developing ePADs is the use of commercial pencil leads as electrodes. Pioneers on using this approach for the fabrication of an ePAD for glucose were Santhiago and Kubota [144] (Figure 5a), who used graphite pencil electrodes as transducers. They used wax printing to define the hydrophobic zones and graphite pencil electrodes as transducers, with a three-dimensional design that was folded to have three separated regions: the first for filtration, the second for the enzymatic reaction and the third for the electrochemical detection. Glucose determination was achieved combining glucose oxidase and 4-aminophenylboronic acid as the mediator, within a range between 0.01 and 1.5 mM, and a limit of detection of 0.38 µM. Pencils can be used as transducers, but they can be also used to draw electrodes onto paper. For example, also in this research group, WitkowskaNery, in collaboration with Santhiago and Kubota [145], developed a paper-based bioactive channel-stacking Whatman no. 3 paper for the detection of glucose and uric acid using a Pt working electrode and a pencil-drawn reference electrode. They performed calibration graphs for glucose from 2 to 10 mM, with a limit of detection of 2 mM (Figure 5b).

Origami strategies for the construction of paper-based analytical devices using pencil-drawn electrodes have been also used by Li et al. [146], who designed a device for the determination of glucose, using chromatographic paper. The linear range and the limit of detection achieved were 1–12 mM and 0.05 mM, respectively, using glucose oxidase and ferrocenecarboxylic acid as the mediator (Figure 5c).

Bernalte et al. [147] critically compared pencil-drawn electrodes with screen-printed electrodes, using different batches of pencils. They concluded that, although their characteristics depend on graphite composition and they are not mass-produced, pencil-drawn electrodes have been shown to be a suitable alternative toward the development of electrochemical paper-based devices.

#### 4.3.4. Drop Casting

Although, as seen before, screen printing was the most commonly used technique for the construction of patterned electrochemical paper-based platforms, there are simpler approaches based on putting a drop of conductive ink on the paper substrate and letting it dry before use, demonstrating its great potential for the development of low-cost glucose biosensors.

With the aim of integrating paper electronics and microfluidics, Hamedi et al. [148] developed a procedure to define (i) microfluidic channels by the well-known wax printing protocol and, after that, (ii) paths for electronic conduction by depositing a water-dispersed conducting polymer mixture of poly(3,4-enthylenedioxythiophene) and polystyrene sulfonate (PEDOT:PSS) or a water dispersion of multiwall carbon nanotubes (MWNTs) and (iii) final electronic/electrofluidic channels by printing wax on part of the already prepared conductive paths. In this case, three different zones can be obtained: one that impedes the flow of solutions, as well as that of electrons (wax-impregnated paper), another that allows only the flow of electrons (paper impregnated with water-dispersed conductive dispersion, and, in a further step, wax) and another, electrofluidic, allowing both the flow of solutions and electrons (paper impregnated with water-dispersed conductive dispersion). In this context, an origami-designed paper-based biosensor was designed for the determination of glucose [148] (Figure 6a). They also developed a paper battery, heater and integrated circuit showing that paper can be a very suitable material to fabricate different electrical devices, thanks to its high surface area, introducing the concept of electrofluidics, referring to areas allowing the simultaneous transport of electrons and fluids. In the application of electrofluidic platforms for glucose determination, they used glucose oxidase as the enzyme and ferricyanide as the mediator.

A wearable paper-based biosensor for the measurement of glucose in sweat was developed by Cho et al. [149] to monitor exercise-induced hypoglycemia. In this device, adhered directly to the skin, sweat is collected by capillary action. It goes to a reservoir where chemical energy is converted into electrochemical energy, on the basis of the electrofluidic structure developed by Hamedi et al. mentioned above [148]. The fuel cell biosensor was made of three layers: the anodic layer, the sweat reservoir, and the air cathode layer, all of them aligned in a Band-Aid^®^ patch. In this way, the anodic reservoir is in contact with human skin, whereas the cathode layer is exposed to the air, due to its cathodic electrochemical activity. The hydrophilic regions were delimited using wax printing, and the conductive reservoir included a mixture of PEDOT:PSS in dimethyl sulfoxide. This makes the paper conductive and serves as a substrate to immobilize glucose oxidase, using a graphene/chitosan solution as the matrix, due to its biocompatibility. The air cathode was fabricated mixing Ni with activated carbon in a binder solution. The wearable biosensor showed a linearity between 0.02 and 1.0 mg/mL, with a sensitivity as high as 1.35 µA·mM^−1^.

A simpler and easier way to construct electrodes on paper is using wax printing for creating hydrophobic barriers and conductive ink drop casting for making the working electrode [41]. Commercially available gold-plated connector headers were used as reference and auxiliary electrodes, and also as connections to the potentiostat. Using an enzymatic cocktail of glucose oxidase and horseradish peroxidase, as well as potassium ferrocyanide as mediator, glucose was determined within concentrations between 0.3 and 15 mM. Using the same design of electrochemical cell, a multiplexed platform (with eight working electrode paper cards) was developed including a glass-fiber tongue to perform sampling without the need of micropipettes [42] (Figure 6b) or diluting, creating real lab-on-paper approaches [87].

Another approach using a paper-based working electrode combined with external reference and auxiliary electrodes, provided by a ceramic SPCE (Screen-Printed Carbon Electrode) card, was developed by Sánchez-Calvo et al. [150], who used glucose oxidase as the enzyme and a cobalt phthalocyanine colloid as the mediator. This sensor determined glucose between 0.1 and 1 mM, with a limit of detection of 63 µM. Another example of using external electrodes was developed by Núnez-Bajo et al. [58], who described the fabrication of an electroanalytical paper-based device with disposable gold-sputtered paper electrodes. The detection zone was limited using a hydrophobic permanent marker and the electrochemical characteristics of the device were evaluated by cyclic voltammetry and differential pulse voltammetry. As a proof of concept, enzymatic determination of glucose was carried out at concentrations between 0.1 and 15 mM, and a limit of detection of 0.11 mM.

### 4.4. Paper as Enzyme Storage

Even though there are many research publications about fabricating electrodes on paper, there are some examples which use a paper-based sensing phase, but external electrodes. Regarding glucose detection, Noiphung et al. [151] (Figure 7a) were able to analyze whole blood samples, exploiting the versatility of paper for integrating steps of the analytical process and designing a µPAD which could separate blood cells from plasma and determine glucose concentration with high selectivity. Microfluidic patterns were created by wax dipping, allowing two different types of paper working together. The detection zone was generated on a commercially available Prussian Blue-modified screen-printed electrode (PB–SPEs) ceramic card. They performed the calibration graph with solutions of glucose concentration ranging between 2.9 and 33.1 mM, before using it with real whole blood samples. The detection of human whole blood has also been achieved by Kong et al. [152] (Figure 7b), using a disposable SPCE combined with a paper disk. The SPCE was modified with a biocomposite made of graphene/polyaniline/Au nanoparticles/glucose oxidase. Electrochemical measurements were carried out by differential pulse voltammetry, measuring concentrations of glucose between 0.2 and 11.2 mM. The limit of detection obtained was 0.1 mM.

SPCEs were also attached to a paper disk by Lawrence et al. [154], using cellulose paper as the storage reagent matrix with the mediator (ferrocene monocarboxylic acid) and glucose oxidase. Glucose was determined using amperometry, within concentrations from 1 to 5 mM, with a limit of detection of 0.18 mM.

Another example is the one published by Dias et al. [153] (Figure 7c). They developed a paper-based enzymatic reactor for batch injection analysis (BIA) of glucose, using a three-dimensional printed cell. After oxidizing the paper surface with sodium periodate, it was activated using N-hydroxysuccinimide (NHS) and N-(3-dimethylaminopropyl)-N’-ethylcarbodiimide hydrochloride (EDC), and then glucose oxidase was covalently immobilized. After adding glucose on the paper-based reactor, which was into a syringe, the reaction took place by vertical flow and the resulting species were collected with a micropipette and analyzed in the BIA cell, containing SPCEs. The device showed a linear behavior between 1 and 10 mM, and a limit of detection of 0.11 mM.

Paper has also been modified with other metals such as platinum in order to construct electrodes with a material different from carbon. Guadarrama-Fernández et al. [155] used platinized paper as support, in conjunction with a biocompatible polymeric membrane containing, among other reagents, glucose oxidase as the biorecognition layer for the detection of hydrogen peroxide generated in the enzymatic reaction. The potentiometric biosensor showed a wide calibration range, between 0.3 and 1 mM, and a limit of detection of 0.02 mM, determining glucose in commercial orange juice samples.

## 5. Conclusions

Nowadays, the demand of valuable information is continuously growing since it is required to make decisions rapidly and accurately. Moreover, if this information is obtained in a rapid and decentralized way, the same could be considered for important health decisions. Instead of having expensive equipment that requires qualified personnel, devices can be moved to the point of need to be used by the final users. In this way, data must only travel from the user to the center to be converted into useful information. In this context, biosensors have attracted much attention due to their excellent characteristics (simplicity, low costs and fast response). Moreover, electrochemical detection improves the features of biosensors, since it allows to obtain quantitative information with simple and portable instrumentation, with high versatility in the way an electrochemical property is measured (e.g., current in (chrono)amperometry/voltammetry or potential in potentiometry). In the biosensing field, glucose enzymatic biosensors are among the most important, considering their relevance in the sensor market and the high amount of published works. The great efforts spent in the development of new designs of glucose biosensors are useful not only to obtain better glucose sensors, but also to pave the way for other less-studied analytes. Although enzymes are being substituted for cheaper materials that are less dependent on reaction media, they continue to be widely used because of their excellent characteristics as biorecognition elements in biosensors (e.g., specificity and sensitivity, among others), allowing them to perform quick and simple assays with very good analytical characteristics. On the other hand, paper is an outstanding material for the development of sensing devices, because of its low cost, disposability, versatility, and ease to be modified with a great number of conductive inks and reagents. In this way, the combination of paper and enzymes has provided a huge number of biosensors and microfluidic platforms for glucose determination. The intention of achieving increasingly cheaper and simpler devices that are yet selective and sensitive, together with the best analytical features of accuracy and precision, will continue, undoubtedly, to produce interesting designs. The development of these novel designs will be focused on achieving increasingly integrated devices able to perform several steps of the analytical process following the “lab-on-a-chip” concept in an autonomous (with minimum human intervention) and decentralized way. In this case, the differences with common ceramic/polymeric glucose sensors will make paper stand out since it is intrinsically flat, it is also flexible, foldable and adaptable, and although it is hydrophilic, it can be modified to become hydrophobic and even omniphobic. Research on new materials with new properties, which are capable of fulfilling different functions at the same time, will be key to achieving those multi-functional integrated devices. However, as used to happen with most of the biosensors developed in the laboratory, the biggest challenge is to fill the gap between the bench and the market. In the case of paper-based analytical devices, to overcome this gap, robust procedures for treating, modifying, folding, etc., the paper are required to be able to manufacture them on a large scale. Although difficult, the research efforts within material science and printed electronics will pave the way for commercial paper-based devices, which, along with the great advances in digital communication networks, will lead to smart tools for analytical applications in any place where they are required.

## Figures and Tables

**Figure 1 sensors-22-06232-f001:**
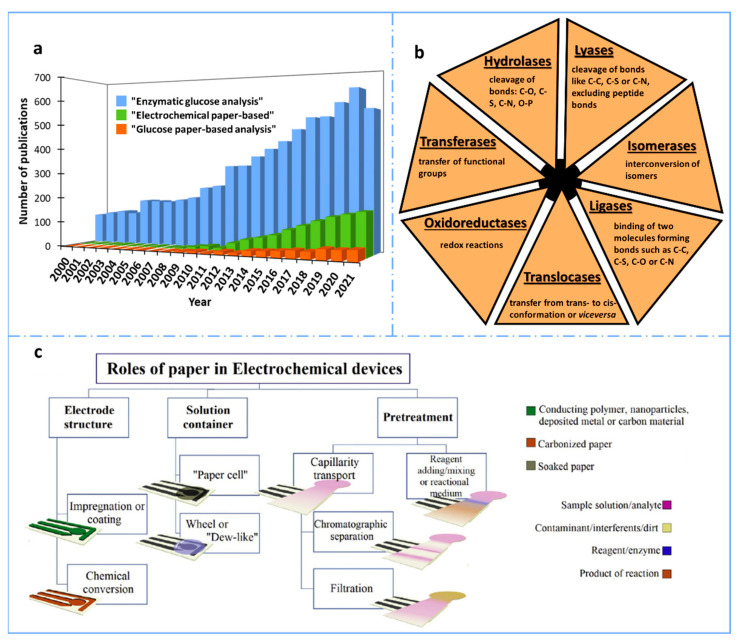
(**a**) Number of results obtained per year when searching “enzymatic glucose analysis”, “electrochemical paper-based” and “glucose paper-based analysis” in the Scopus database for the last 22 years (2000–2021). (**b**) Classification of the enzymes according to the International Union of Biochemistry and Molecular Biology [11]. (**c**) Schematic representation of the main roles of paper in electroanalytical devices. Reprinted from [12], Copyright (2020), with permission from Elsevier.

**Figure 3 sensors-22-06232-f003:**
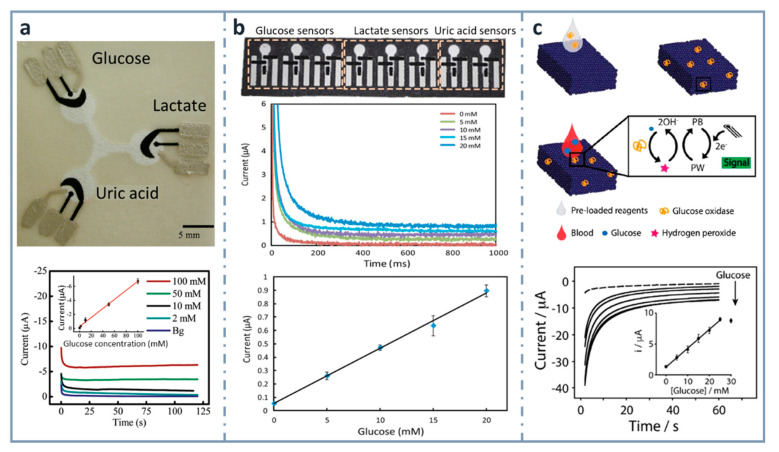
Screen-printed, paper-based biosensors. (**a**) Picture of the first three-electrode paper-based device for the determination of glucose, lactate and uric acid (top) and chronoamperograms for glucose with the calibration plot in the inset (bottom). Reprinted from [90], Copyright (2009), American Chemical Society. (**b**) Paper-based electrochemical biosensor array (top), chronoamperometric curves (center) and calibration plot for glucose in artificial urine (bottom). Reprinted from [134] (open access). (**c**) Measurement scheme of the glucose biosensor on Paper Blue (top), chronoamperometric detection of increasing concentrations of glucose with calibration curve of glucose up to 30 mM in the inset (bottom). Reprinted from [135], Copyright (2018), with permission from Elsevier.

**Figure 4 sensors-22-06232-f004:**
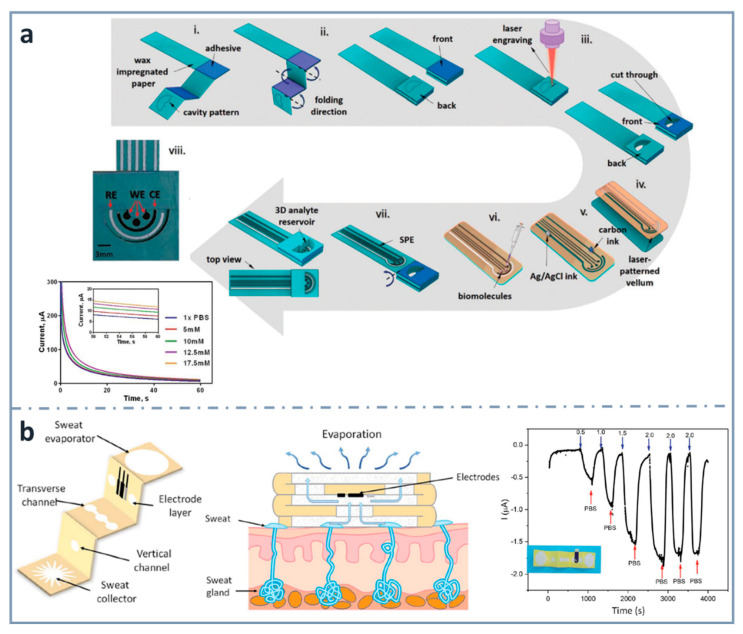
Screen-printed three-dimensional paper-based biosensors. (**a**) Fabrication process of the origami device for the detection of glucose and corresponding chronoamperometric signals (bottom left). Reprinted from [139], Copyright (2018), with permission from The Royal Society of Chemistry. (**b**) Schematic diagram of the wearable biosensor (left) for determining glucose in sweat (center) and amperometric response for glucose (right). Reprinted from [140] (open access).

**Figure 5 sensors-22-06232-f005:**
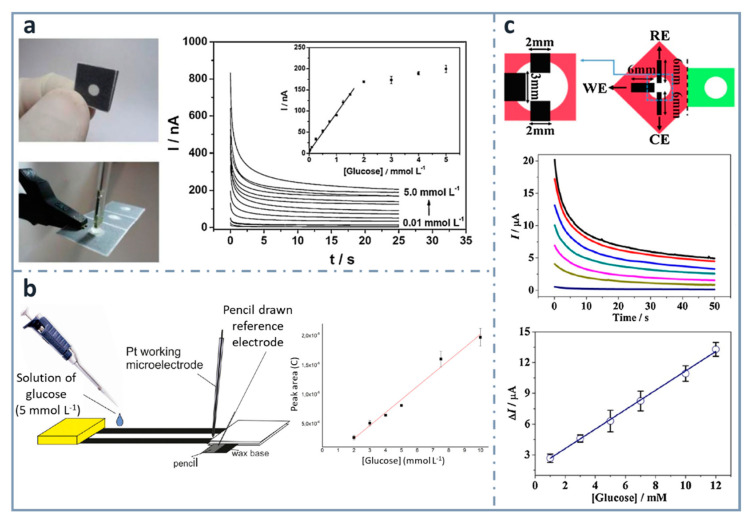
Pencil-based electrodes on paper-based devices for glucose detection. (**a**) Folded and unfolded device with a two-electrode system made of a graphite pencil as the working electrode and silver ink as the reference/auxiliary electrode (left). The chronoamperometric response of increasing concentrations of glucose is included with the calibration curve in the inset (right). Reprinted from [144], Copyright (2013), with permission from Elsevier. (**b**) Layout of the paper-based device with a pencil-drawn reference electrode (left) and calibration curve for glucose (right). Reprinted from [145], Copyright (2016), with permission from Wiley. (**c**) Three-electrode system drawn using a graphite pencil (top), chronoamperograms (center) and calibration graph for glucose detection (bottom). Reprinted from [146], Copyright (2016), with permission from Elsevier.

**Figure 6 sensors-22-06232-f006:**
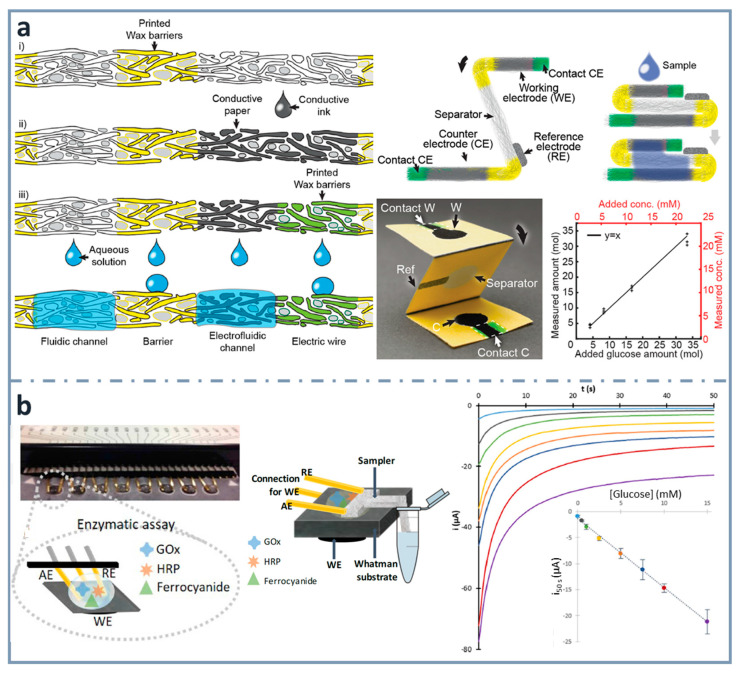
(**a**) Schematics of the fabrication of th electrofluidic device using wax-printed barriers and conductive ink-dropped electrodes (left), assembly of the electroanalytical device (top right and picture at the center) and glucose calibration graph (bottom right). Reprinted from [148], Copyright (2016), with permission from Wiley. (**b**) Multiplexed paper-based enzymatic array of glucose biosensors (left), combined with a sampler for the collection of the sample from the container (center) and corresponding chronoamperograms, with the calibration graph for glucose in the inset (right). Reprinted from [42], Copyright (2019), with permission from Elsevier.

**Figure 7 sensors-22-06232-f007:**
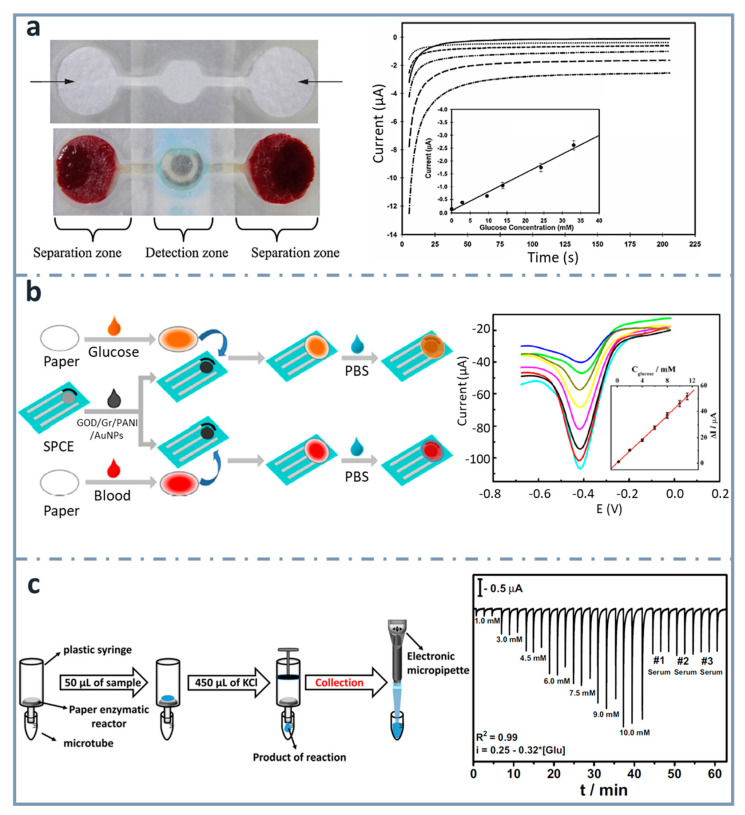
ePADs using external electrodes. (**a**) Picture of dumbbell-shaped ePAD for whole blood separation and ePAD assembled with the SPCE (top and bottom left, respectively). Chronoamperograms corresponding to the glucose assay are also shown, with the calibration curve as an inset (right). Reprinted from [151], Copyright (2013),with permission from Elsevier. (**b**) Schematic representation of the fabrication and assay procedure of the paper-based glucose sensor (left) and differential pulse voltammetry signals of the device with the calibration curve as an inset (right). Reprinted from [152], Copyright (2014), with permission from Elsevier. (**c**) Scheme showing the enzymatic assay performed on the paper-based reactors, the product of which is analyzed in the 3D-printed BIA cell couple with the screen-printed electrodes (left), and amperometric response for increasing glucose concentrations (right). Reprinted from [153], Copyright (2016), with permission from Elsevier.

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
