# Peer review of "Paper-Based Enzymatic Electrochemical Sensors for Glucose Determination"

_sensors, 2022, doi:10.3390/s22166232_

Round 1

Reviewer 1 Report

I think it’s an interesting, important, and timely Review. I would recommend it after a minor revision.

1. Make a summary about what kinds of papers are used in electrochemical sensors.

2. Highlight the strategies how the paper-based electrochemical sensors tackle the issue of electrical conductivity.

3. Provide one Figure about the electron transfer pathway between enzyme in sensor and glucose.

4. Pay attention to some usages of nanomaterials onto paper-based sensors, such as MOFs-derived nanomaterials, hollow nanoporous carbon framework, carbon nanotubes.

5. Many research works have been done on the paper-based enzymatic electrochemical sensors for glucose detection. Please give some possible directions for the next-stage research.

Reviewer 2 Report

Overall a very comprehensive review.  However, it covers many more papers than just those published in the last five years (as stated in the Abstract) on paper enzymatic electrochemical glucose biosensors, reading like the Introduction to a Thesis perhaps.  Much of it comprises generic information on materials, enzymes, and fabrication of devices, not necessarily on the latest paper electrochemical biosensors for glucose. Depending on the requirements of the Editor, this may or may not be desirable, since for the expert in the field, there is a lot of content to read through before being able to focus directly on the latest papers on paper-based electrochemical biosensors for glucose per se.  The section on enzymes and that on immobilisation techniques could arguably be removed.  However, if read in full, the article does progress logically and for some readers the background information will add useful context.

Hence, in the comments to be addressed below, I have only recommended minor alteration to remove a short section of text, together with some other amendments, mainly re-wording for clarity, or grammatical in nature.  

Please see the comments below:

11)      Some repetition and less relevant material can be removed, also having the effect of shortening the review:  I suggest removing lines 495 (“Nowadays…”) – 505 (“…electrochemical [115]. In special”).  Begin the paragraph with “Colorimetric detection is very appropriate…”

22)      Re-wording/grammatical amendments for clarity:

Line 12: replace “notorious efforts on improving” with something else (Notorious suggests bad reputation!). One suggestion would be ”…has made continuous progress in improving…”

Line 31: as an energy reserve…

Line 62: I suggest beginning the sentence with “In terms of material, one of the strategies…”

Line 64: replace “notorious” with “dramatic”.  (Notorious suggests bad reputation!).

Line 64: in the last ten years

Line 68: …converts it into an environmentally friendly material, among others.

Line 72: in the last recent years.

Line 76: landmarks which whose origin…

Line 84: …, the paper…

Line 85/86: “In 1956, the first latex agglutination assay was developed by Plotz and Singer,…”

Line 89/90: …LFAs has increased…

Line 93: Whitesides and col   et al

Line 98: , it was Whitesides’ work…

Line 100: experimented experienced

Line 104: associated to its with their low cost…

Line 109: makes this material…

Line 115: of   or

Line 123: papers employed as…

Line 157: flammable it is useful…

Line 161/162: …makes residue removal difficult, and makes…

Line 165: beard borne

Line 169: pay play

Line 178: very simple, what makes it making it portable…

Line 178 - 179: “All this, maintaining…” THIS SENTENCE DOES NOT MAKE SENSE - NEEDS REWORDING OR REMOVING. A SUGGESTION WOULD BE TO INSERT “sensitivity and selectivity of” into the next sentence before “electrochemical…”

Line 204: and col  et al

Line 205: employ   use

Line 212: in   the last   recent years

Line 218: …between amongst others.

Line 255: REWORD THIS: “In 2018, a seventh group of enzymes was discovered [67] – translocases, that…”

Line 259: in the last recent years

Line 259: is attracting   has attracted

Line 260: homologue   homologous

Line 376: oxidase   oxidises

Line 380-381: it is common the to use auxiliary reactions…

Line 429: flows

Line 535: are being described

Line 737: “…within concentrations comprised between 0.3 and 15 mM.”

Line 744: Insert the reference number = 118(?) after the author name as follows: ”…by Sanchez-Calvo et al [118],… “

Line 766: approaching   exploiting

Line 811: properly?  Not sure what you mean here. Rapidly? Correctly? “rapidly and accurately”?

Line 813: requires

Line 816: “…attracted  a  great attention due to its their excellent characteristics…”

Line 829: “…, paper has shown to be is an outstanding material…”

33)      Other points to be clarified/checked:

Line 131: “basis weight” – is this the correct term? Or do the authors mean basic weight? Is this a recognised term? Please check.

---------------------------------------------------------------------------------------------------------------------
